# Study on the Unconventional Water Subsidy Policy in the Arid Area of Northwest China

Chaomeng Ma, Hongzhen Ni *, Yunzhong Jiang and Xichen Lin

State Key Laboratory of Simulation and Regulation of Water Cycle in River Basin, China Institute of Water Resources and Hydropower Research (IWHR), Beijing 100038, China
* Correspondence: nhz6969@iwhr.com; Tel.: +86-010-6878-5702

**Abstract:** The arid regions of Northwest China are facing water shortages and ecological fragility. Making full use of unconventional water is one of the effective ways of solving water issues and achieving high-quality regional development. The high cost of unconventional water utilization is the main obstacle to its utilization and technological development, and the subsidy policy may become a breaking point. Taking Ningdong Energy and Chemical Industry Base (NECI Base) as a case study, the article proposes raising the Yellow River water price to subsidize the utilization of mine water. The development and utilization of mine water can be effectively improved. Considering the optimal allocation of multiple water sources and the substitution relationship between the Yellow river water and mine water, this paper extends the water resources module (WRM) of the Computable General Equilibrium (CGE) model. The model can reflect the substitution of water sources and the linkage between water prices and the economy. Ten different subsidy policy scenarios are simulated through the extended CGE model, and the laws and mechanisms of the subsidy policy on the economy and water usage are summarized. The results show that increasing the price of Yellow River water by 8% to subsidize the mine water will achieve optimal socio-economic output. Under this scenario, the industrial value added (IVA) is basically unaffected, the water-use efficiency (WUE) is significantly improved, and the affordability of the enterprise is satisfied. The Yellow River water usage decreased from 319.03 million (M)m³ to 283.58 Mm³ (11.1% saving), and mine water usage increased from 27.88 Mm³ to 47.15 Mm³ (69.1% increase).

**Keywords:** mine water; arid area; water price; subsidy; policy; CGE model; water pressure

## 1. Introduction

With global climate change, the water shortage has become one of the main factors restricting human economic and social development [1,2]. Unconventional water resources, also known as "second water sources" for economic and social development, play an important role in alleviating urban water shortages, controlling environmental pollution, and achieving high-quality development [3–5]. Different from conventional water sources such as river water and groundwater, unconventional water resources mainly refer to the utilization of reclaimed water, salt water, mine water, rainwater, seawater, and floods [6]. Despite strong government advocacy and support in China, the development and utilization of unconventional water are still in their early stages. Since the implementation of the "Strictest Water Resources Management System" in 2012, the utilization of unconventional water in China has accelerated (1.74% of the total water supply in 2020). However, there is still a big gap compared with Singapore, Israel, Portugal, and other countries [6]. From the perspective of spatial distribution, the utilization rate of unconventional water is relatively high in the northwest and northeast of China, while it is relatively low in the center, south, and east of China.

In the water usage structure of the Ningdong Energy and Chemical Industry Base (NECI Base), unconventional water sources mainly come from two aspects: reclaimed water

and mine water. At present, the utilization rate of reclaimed water is close to 100%, while the utilization rate of mine water is still insufficient (about 30%). A large amount of unused mine water is discharged into the river and lake system, which causes environmental pollution (the mine water has high salinity) [7–9]. One of the main reasons for the low utilization rate of mine water is that the utilization cost is much higher than that of the Yellow River water. Therefore, finding reasonable policy incentives and optimizing water resource allocation is of great significance for improving the water-use efficiency (WUE) of unconventional water.

The research literature on water resource allocation was first seen in the 1960s. From the 1960s to the early 21st century, water resource problems occurred frequently, the water environment gradually deteriorated, and water resource shortage became a major social problem. Dudley et al. [10] use the relevant model to reasonably allocate irrigation water resources. Wong et al. [11] propose a multi-objective method for the optimal allocation of water resources and introduce the principle and advantages of this management method. In the 21st century, the problems of water shortage and serious water pollution are particularly prominent. The previous water resource management model is no longer suitable for the current water resource field, and the research on the optimal allocation of water resources enters the development stage. Considering the social, economic, and environmental aspects, Dedi Liu et al. [12] build the model for the optimal allocation of water resources in a saltwater intrusion area and adopt a genetic algorithm to optimize the model. Amar Razzaq et al. [13] prove that the informal groundwater markets improve WUE and equity by conducting a field survey of 120 farmers. Amar Razzaq et al. [14] indicate that policy intervention of standardized groundwater marketing contracts can help reduce the overexploitation of groundwater and environmental externalities. At the same time, research on the optimal allocation of water resources in China began to develop at a considerable speed. Xinyi Xu, Hao Wang [15], Xinmin Xie [16], and others compile regional water allocation theories and related strategies in North China and Ningxia. Xiao Hu [17] studies the multi-objective optimal allocation of water resources in the Daxia River basin based on the genetic algorithm optimization of a large-scale system in general. Xiaoxiang Zhang et al. [18] use the compromise programming algorithm to solve the optimal allocation model of water resources in the Fuhe River basin.

At present, the commonly used water resource allocation models include the game model, multi-objective optimization model, input and output optimization model, and computable general equilibrium (CGE) model. The water resource allocation game model considers the overall benefits and the balance of water supply and demand, and it conducts the primary and secondary water allocation games among different water users [19,20]. The multi-objective optimization model generally aims to maximize the economic, social, and ecological benefits or minimize costs and losses, and it is usually solved by intelligent optimization algorithms [21]. The input and output optimization model is mainly used to study the optimal scheme to realize the multi-objective optimal allocation of water resources, economy, and environment [22]. The CGE model has been widely used in policy research on resources, the environment, and trade [23].

Research on unconventional water resource utilization has achieved many advancements, which can be roughly divided into three categories according to the research direction: policy and theoretical research, qualitative analysis, and economic benefit analysis of unconventional water utilization. As for policy and theoretical research of unconventional water utilization: Yi et al. [24] comment on the history, current situation, potential, relevant regulations and policies, and risks of reclaimed water utilization in order to ensure the proper operation of water utilities. F. Silva Pinto and R. Cunha [25] study objectives that need to be assessed for setting water tariffs. The objectives include economic efficiency, financial sustainability, environmental concerns, social concerns, and governance. Bick et al. and Chang et al. [26,27] conducted a specific analysis of the key policies and factors affecting the utilization of reclaimed water. D. Bixio et al. [28] propose clear mechanisms, subsidy policies, and standards and guidelines for promoting unconventional water utilization.

Based on the policy and theoretical research of unconventional water utilization, there are many studies focusing on the qualitative analysis of unconventional water utilization [29]: Lihong Lv [30] qualitatively analyzes the economic and social benefits of reclaimed water utilization. Yushan Wan [31] analyzes the direct and indirect comprehensive benefits of reclaimed water to the city. The economic benefit analysis of unconventional water utilization mainly focuses on theoretical research and qualitative analysis, with few on quantitative analysis: Nan Xiang [32] studies the impact of reclaimed water utilization on the economy and environment in Tianjin by improving the water resources–environment–economy model.

The research on unconventional water utilization mainly focuses on theoretical and qualitative analysis, with less quantitative analysis. There is more research on technical analysis or cost–benefit analysis for quantitative analysis, with less research on comprehensive and systematic policy evaluation and analysis. As for the policy and theoretical research, many scholars consider promoting unconventional water utilization from the perspective of unconventional water or conventional water unilaterally. There is less research on the linkage mechanism between unconventional water and conventional water [4,33,34]. Based on the insufficiency of the above unconventional policy research, in order to promote the utilization of mine water, this paper proposes raising the Yellow River water price to subsidize mine water (hereinafter referred to as the mine water subsidy policy). The mine water subsidy policy establishes the price linkage relationship between the Yellow River water and mine water, and it can promote mine water utilization with less economic fluctuations.

In view of the research status of less quantitative analysis in unconventional water utilization, this paper chooses the CGE model to quantitatively analyze the mine water subsidy policy. The CGE model reflects overall economic linkage and change in price to demand and output, and it has advantages in the quantitative analysis of the unconventional water utilization policy [35–37]. Compared with other research methods, the CGE model has the following advantages when analyzing the unconventional water subsidy policy: (i) It can reflect the impact and feedback process of the policy on industrial sectors, residents, governments, investors, and other economic entities. (ii) It can describe the supply–demand balance of water resources, thus reflecting the impact of the policy changes on the supply and demand sectors of water resources. (iii) It can describe how the policy changes affect the prices of labor, capital, and other primary factors, thus acting on the production sector and the macro-economy. More importantly, to study the mine water subsidy policy, this paper extends the water resource module (WRM) of the CGE model based on considering the optimal allocation of water resources in the NECI Base. The extended CGE model can reflect the substitution relationship between the Yellow river water and mine water, and it can embody the optimal allocation of water resources in the NECI Base. In addition, the extended CGE model can analyze the impact of the price of the Yellow River water and mine water on the macro-economy, and it can reflect the impact of the price of one water source on other water sources' usage.

After this brief introduction, Section 2 introduces the general situation of the study area and the research method. The study results and discussion of the results are presented in Section 3. Finally, Section 4 draws some concluding remarks and provides some recommendations to the authorities.

## 2. Materials and Methods

### 2.1. Study Area

The NECI Base, located in the arid area of northwest China, is a large-scale coal production base in China. It is an important engine for the economic and social development of the Ningxia Hui Autonomous Region (NHAR). Its industrial value added (IVA) accounts for about one-third of the NHAR, as shown in Figure 1. The Yellow River water is the main water source in the NECI Base. In order to promote sustainable development along the Yellow River, the ecological protection and high-quality development of the

Yellow River was incorporated into the national strategy in 2020. In response to the above, NHAR sets the "red line" (the control index of the total water usage) of the NECI Base at 200 million (M)m³. However, in 2020, the actual total water usage of NECI Base was about to break through the "red line", and water-saving measures became imperative. Therefore, making full use of unconventional water is the key to solving water shortages and achieving high-quality development.

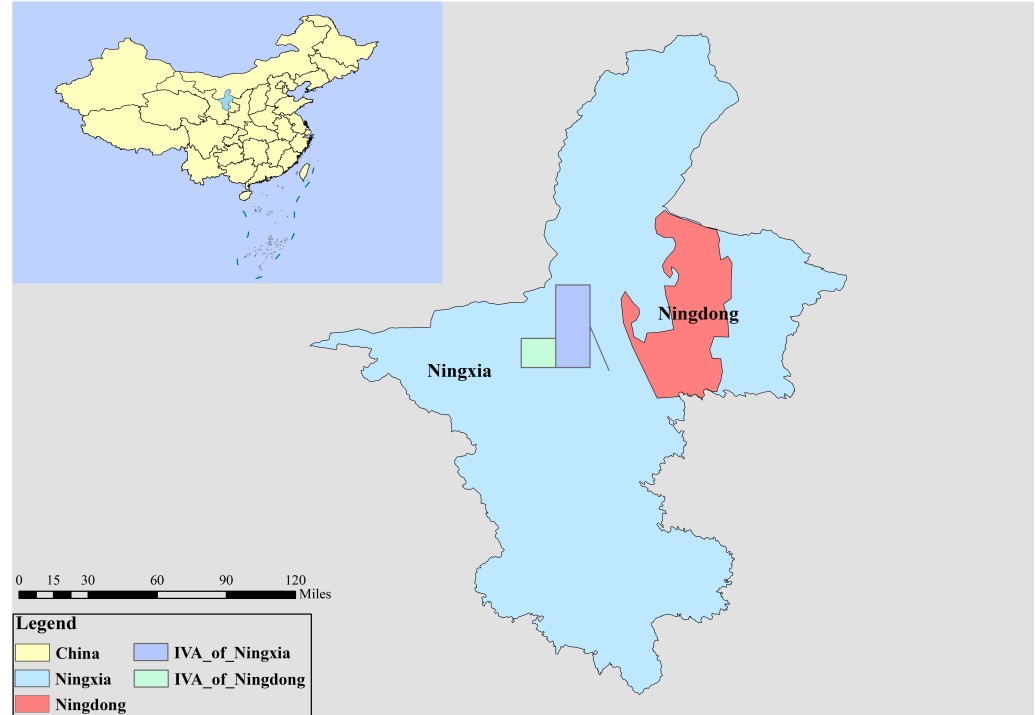

**Figure 1.** Location of NECI Base (Sources: Survey of NECI Base).

### 2.1.1. Water Usage

The water sources used by the NECI Base include the Yellow River water, unconventional water, and groundwater. The total water usage is increasing year by year, as shown in Table 1. The Yellow River water usage is the most among all water sources, with an average annual usage of about 180 Mm³. Unconventional water includes reclaimed water and mine water. The reclaimed water usage grows more and more year by year, with a nearly 100% utilization rate at present. Mine water usage does not change much, with only a 30% utilization rate; most of the mine water is discharged to Nan Lake. The groundwater is basically domestic water, with a very little usage.

**Table 1.** Water usage of different water sources.

| Year | The Yellow River | | Unconventional Water | | | | Groundwater | | Total |
| --- | --- | --- | --- | --- | --- | --- | --- | --- | --- |
| | Usage | Rate | Reclaimed Water Usage | Mine Water Usage | Subtotal | Rate | Usage | Rate | |
| | (Mm³) | | (Mm³) | Mm³ | (Mm³) | | (Mm³) | | Mm³ |
| 2017 | 187 | 81.60% | 29 | 10 | 39 | 16.80% | 4 | 1.50% | 229 |
| 2018 | 174 | 76.70% | 36 | 13 | 48 | 21.30% | 5 | 2.00% | 227 |
| 2019 | 172 | 63.30% | 75 | 20 | 95 | 35.10% | 4 | 1.60% | 272 |
| 2020 | 179 | 64.60% | 76 | 19 | 96 | 34.50% | 3 | 0.90% | 277 |

Sources: Survey and Statistics of NECI Base.

The water usage of different industries is shown in Table 2. Industrial water usage is the largest, accounting for more than 90% of the total water. Domestic and green water usage is relatively small, accounting for less than 10% of the total water.

**Table 2.** Water usage of different industries.

| Year | Industrial Water | | Domestic Water | | Green Water | | Total (Mm³) |
|---|---|---|---|---|---|---|---|
| | Usage (Mm³) | Rate | Usage (Mm³) | Rate | Usage (Mm³) | Rate | |
| 2017 | 211 | 92.40% | 2 | 0.80% | 16 | 6.80% | 229 |
| 2018 | 215 | 95.00% | 2 | 0.70% | 17 | 7.50% | 227 |
| 2019 | 254 | 93.60% | 2 | 0.80% | 21 | 7.60% | 272 |
| 2020 | 254 | 91.80% | 2 | 0.90% | 25 | 8.90% | 277 |

Sources: Survey and Statistics of NECI Base.

### 2.1.2. Water Cost

The Yellow River water is mainly supplied by the Ningdong Water Supply Project at 2.8 Chinese Yuan (CNY)/m³. The operating cost of the mine water treatment plant is generally around 10.0 CNY/m³, and the price of mine water supplied to the surrounding power plants is 10.8 CNY/m³. The mine water cost is much higher than the Yellow River water price, which is the main reason for the low utilization efficiency of mine water.

### 2.1.3. Affordability

The industry affordability is mainly analyzed according to the proportion of industrial water cost to the IVA. According to the research of the World Bank and some international lending institutions, when the water fee accounts for 3% of the IVA, the industry will pay attention to water usage [38]. Considering the actual situation of the industry in China, 2% of the IVA is selected as the analysis standard.

This paper collects the IVA and water usage of representative enterprises in the NECI Base. According to the above analysis, it is calculated that the affordability of the industry in the NECI Base is 8.78 CNY/m³.

### 2.1.4. Water Resources Planning

According to the "Water Resources Allocation of NECI Base during 14th Five-Year Plan (2021–2025)", the plan is to allocate the Yellow River water 285.12 Mm³, allocate the mine water 48.68 Mm³, and allocate the reclaimed water 9.01 Mm³ by 2025. The water resource allocation can meet the restriction of the total water usage and the maximum utilization of mine water. It can relieve the pressure on the Yellow River water, and the negative impact on the ecological environment. The mine water is allocated in power plants and coal chemical industry parks but not in new material industries and other industries. The specific water resource allocation is shown in Table 3.

### 2.2. Methods

### 2.2.1. ORANI-G Model

The ORANI-G model is a multi-sectoral model developed by the Center of Policy Studies (CoPS) at the University of Victoria, Australia, based on neoclassical economic theory. The model can distinguish two different input sources as local or imported. It includes detailed industrial sector classification, multiple primary factor inputs, and rich economic agents [39]. The ORANI-G model can describe the economic reality in detail. It can be used to simulate the impact of resource and environmental policy changes on the resources, environment, industrial activities, and macro-economics [40,41].

**Table 3.** Water allocation in 2025.

| Industries | | Water Demand (Mm³) | Water Allocation (Mm³) | | |
|---|---|---|---|---|---|
| | | | Public Reclaimed Water | Mine Water | The Yellow River Water |
| Industry | Power plants | 42.95 | | 11.57 | 31.38 |
| | Coal mine | 29.97 | | 24.98 | 5 |
| | Coal chemical industrial park | 112.18 | | 12.14 | 100.04 |
| | Linhe Industrial Park | 70.45 | 0.61 | | 69.85 |
| | New material chemical industry park | 38.57 | 3.4 | | 35.17 |
| | International Chemical Industry Park | 1.9 | | | 1.9 |
| | Other industries | 23.83 | | | 23.83 |
| | Subtotal | 319.85 | 4.01 | 48.68 | 267.16 |
| Domestic water | | 2.96 | | | 2.96 |
| Green water | | 20 | 5 | | 15 |
| Total | | 342.81 | 9.01 | 48.68 | 285.12 |

Sources: Water Resources Allocation of NECI Base during the 14th Five-Year Plan (2021–2025).

The ORANI-G model includes six economic agents: production, investment, residents, government, foreign countries, and inventory. It constructs a behavioral mechanism equation for each economic agent based on economic theory. The production agent uses the multi-layer nested Constant Elasticity of Substitution (CES) function to describe the producer's cost minimization behavior. The CES function determines the producer's choice of intermediate inputs and primary factor inputs. The investment agent determines the allocation of investment according to the expected return on investment, it uses the CES function to determine the best investment portfolio. The resident sector uses the Klein–Rubin utility function to describe the behavior of residents' utility maximization to determine the choices of commodities, which is usually set as an exogenous variable. To simplify the analysis, the government agent is assumed to have the same proportion of change as resident consumption expenditure.

The ORANI-G model is mainly used to construct the national multisectoral CGE model in the early stage. Based on the theoretical framework of the ORANI-G model, Deng Qun et al. [42], Xia et al. [43], and Dixon et al. [39] also developed a single-province or single-region multi-sector CGE model. The traditional national CGE model has the same theoretical mechanism and a similar equation form as the single-region CGE model. The difference is that the import and export commodities of the single-province or single-region CGE model include not only the commodities imported and exported from abroad but also from other domestic regions.

### 2.2.2. Main Module

Based on the single-region CGE model, this paper extends WRM to study the impact of the mine water subsidy policiy on water usage and the economy. The study is closely related to the production behavior of the industrial sector. Therefore, this paper mainly introduces the production module of the ORANI-G model and the WRM; the model structure is shown in Figure 2.

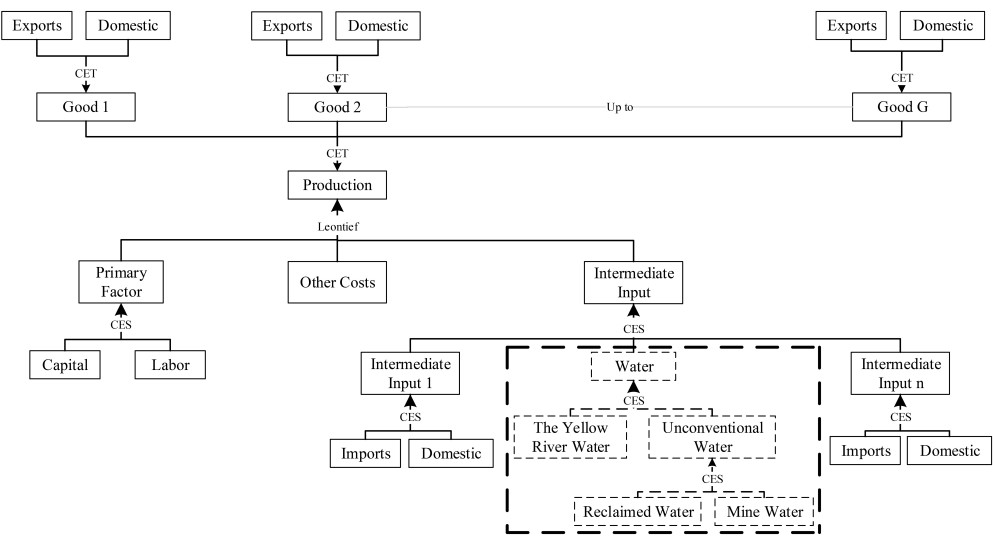

**Figure 2.** Model structure (adapted from [39]).

1. Production Module

In the ORANI-G model, producers use primary factors and intermediate inputs for production. In terms of inputs, the production process is described with a multi-layer nested CES function. Each layer of nesting is assumed to be an independent production process, and the optimal combination of inputs in each nesting is not directly related to the price of other nesting. In terms of output, the production has two different uses: local and export. The Constant Elasticity of Transform (CET) function is often used to describe the output process. The solid line in Figure 2 shows the input–output structure of the production module in the ORANI-G model.

In the first layer of nesting, intermediate inputs $X1\_S(c, i)$, primary factors $X1PRIM(i)$, and other costs $X1OCT(i)$ required in the production process are nested through the Leontief production function, as shown in Equation (1) [39].

$$X1TOT(i) = \frac{1}{A1TOT(i)} MIN\left[All, c, COM : \frac{X1\_S(c,i)}{A1\_S(c,i)}, \frac{X1PRIM(i)}{A1PRIM(i)}, \frac{X1OCT(i)}{A1OCT(i)}\right] \quad (1)$$

where $A1TOT(i)$ represents the Hicks-neutral technical change term, and its change has the same effect on $A1LAB(i)$ and $A1CAP(i)$.

In the second layer of nesting, primary factors are disassembled into labor $X1LAB(i)$ and capital $X1CAP(i)$ through the CES production function, as shown in Equation (2) [39].

$$X1PRIM(i) = CES\left[\frac{X1LAB(i)}{A1LAB(i)}, \frac{X1CAP(i)}{A1CAP(i)}\right] \quad (2)$$

At the same time, the intermediate inputs $X1_S(c, i)$ are disassembled into the domestic input and imported input through the CES function, as shown in Equation (3) [39]. Based on the Armington hypothesis [44], the use combination of the two commodities in the intermediate use process is set by the cost minimization function.

$$X1\_S(c, i) = CES\left[ALL, S, SRC : \frac{X1(c,s,i)}{A1(c,s,i)}\right] \quad (3)$$

2. WRM

The input–output structure of the ORANI-G model can also be used to describe the water production and supply industry; the water goods are output products of the water production and supply industry. Water goods are intermediate inputs in all industrial sectors, so they can be incorporated into the production process as intermediate inputs in this paper. Combined with the water structure in the study area, the "water" is subdivided

into "the Yellow River water", "reclaimed water", and "mine water". According to the strength and possibility of the water source substitution relationship, a multi-layer nesting method is adopted to extend the WRM. The logical structure is shown in the dotted box in Figure 2.

With the rapid development of the unconventional water industry, unconventional water can replace the Yellow River water in some industries and uses. Therefore, water goods $X_{water,i}^{(1)}$ is disassembled into unconventional water and the Yellow River water by the CES function, as shown in Equation (4).

$$X_{water,i}^{(1)} = CES\left\{ \frac{X_{(water,s),i}^{(1)}}{A_{(water,s),i}^{(1)}}; \rho_{water,i}^{(1)}, b_{(water,s),i}^{(1)} \right\}(i = 1,\ldots,n; s = 1,2) \tag{4}$$

where $X_{(water,s),i}^{(1)}$, $A_{(water,s),i}^{(1)}$, and $b_{(water,s),i}^{(1)}$ represent, respectively, the input amount, technical parameters, and share parameters of water goods from the source ($s = 1$ means Yellow River water; $s = 2$ means unconventional water) to the industry sector $i$, and $\rho_{water,i)}^{(1)}$ is the constant substitution elasticity coefficient of composite water goods in an industry $i$.

It is believed that reclaimed water and mine water have a substitution relationship and have a certain elasticity of substitution, so the unconventional water $X_{uncon\_wat,i}^{(1)}$ is disassembled into reclaimed water and mine water through the CES function, as shown in Equation (5).

$$X_{uncon\_wat,i}^{(1)} = CES\left\{ \frac{X_{(uncon_wat,s),i}^{(1)}}{(A_{(uncon_wat,s),i}^{(1)}}; \rho_{uncon_wat,i}^{(1)}, b_{(uncon_wat,s),i}^{(1)} \right\}(i = 1,\ldots,n; s = 1,2) \tag{5}$$

where $X_{(uncon\_wat,s),i}^{(1)}$, $A_{(uncon\_wat,s),i}^{(1)}$, and $b_{(uncon\_wat,s),i}^{(1)}$ represent, respectively, the input amount, technical parameters, and share parameters of composite commodities from the source ($s = 1$ means reclaimed water; $s = 2$ means mine water) to the industry sector $i$.

### 2.2.3. Database

Collecting the economic and water usage data of the NECI Base and combining the data with the input–output table of NHAR, the CGE model for the NECI Base can be established. The model parameters were determined through investigation and previous research.

For economic data, based on the input–output table of 42 industries in NHAR in 2017, combined with the industry planning of the NECI Base, the input–output table of 42 industries is merged into 13 industries, as shown in Table 4. Based on the actual economic data of NECI Base, the input–output table of 13 industries in NHAR in 2017 is updated to the input–output table of 13 industries in the NECI Base in 2020. The RAS method is used to balance the input–output table of 13 industries of the NECI Base in 2020.

For water utilization data, the actual industrial water usage of the NECI Base in 2020 is collected. The water usage of "Yellow River water", "reclaimed water", and "mine water" is classified according to the industry classification in Table 4. Then, the water structure of different sources and industries in the NECI Base can be obtained.

**Table 4.** Model sectors merged from 42 sectors of the national economy industry classification.

| Sector Classification | 13 Sectors | 42 Sectors Based on the National Economy Industry Classification |
|---|---|---|
| Primary Sector | Agriculture | Agriculture, Forestry, Animal Husbandry, and Fishery |
| Secondary Sector | Coal mining and chemical industry | Mining and washing of coal; Extraction of petroleum and natural gas; Mining and processing of metal ores; Mining and processing of nonmetal and other ores; Processing of petroleum, coking, processing of nuclear fuel; Manufacture of chemical products; Production and distribution of gas |
| | Thermal power industry | Production and distribution of electric power and heat power |
| | New materials industry | Manufacture of chemical products |
| | Other industry | Food and tobacco processing; Textile industry; Manufacture of leather, fur, feather, and related products; Processing of timber and furniture; Manufacture of paper, printing and articles for culture, education, and sport activity; Manufacture of non-metallic mineral products; Smelting and processing of metals; Manufacture of metal products; Manufacture of general-purpose machinery; Manufacture of special-purpose machinery; Manufacture of transport equipment; Manufacture of electrical machinery and equipment; Manufacture of communication equipment, computers, and other electronic equipment; Manufacture of measuring instruments; Other manufacturing; Repair of metal products, machinery, and equipment; Comprehensive use of waste resources |
| | Yellow River water supply industry | Water production and supply industry |
| | Reclaimed water supply industry | Water production and supply industry |
| | Mine water supply industry | Water production and supply industry |
| Tertiary Sector | Construction | Construction |
| | Trade | Wholesale and retail trades |
| | Transport | Transport, storage, and postal services |
| | General service | Administration of water, environment, and public facilities; Finance; Information transfer, software, and information technology services; Leasing and commercial services; Real estate; Scientific research and polytechnic services |
| | Water intensive service | Accommodation and catering; Culture, sports, and entertainment; Education; Health care and social work; Public administration, social insurance, and social organizations; Resident, repair, and other services |

Sources: Adapted from [45].

According to the actual situation of the study area and the research results of references, the model parameters are assigned: (1) The elasticity coefficient in labor demand is estimated by the Chinese Academy of Social Sciences at 0.243 [46]. (2) Consumer price elasticity is assigned 4 from the People's Republic of China general equilibrium model (PRCGEM) [47]. (3) Industry Armington elasticity, factor substitution elasticity, and resident consumption elasticity adopt the results of the ORANI-G model. (4) According to the references' research results and the regional per capita income level, the Frisch parameter is assigned −2 [48]. (5) Other parameters of the model use ORANI-G model parameters.

### 2.3. Policy Scenarios

According to different stages of unconventional water, unconventional water subsidies are divided into three aspects in China: research and development (R&D), construction, and operation [4]. The forms of subsidies include rewards instead of subsidies, direct financial subsidies, and price support. The subsidy subject of the R&D stage and the construction stage is mainly the central financial authorities, and the subsidy subject of the operation stage is mainly the local finance sector. (i) Subsidies for the R&D stage are mainly used for research projects and the R&D of unconventional water treatment equipment. The forms of subsidies include direct and indirect financial subsidies. (ii) Subsidies for the construction stage are mainly used for the construction, repair, and reconstruction of unconventional water projects. The forms of subsidies include direct financial subsidies and rewards instead of subsidies. (iii) Subsidies for the operation stage are mainly used to reduce the unconventional water price owing to the high cost of unconventional water treatment. The forms of subsidies include direct financial subsidies and r price support. Considering the actual situation of the unconventional water treatment scale and unconventional water volume in the NECI Base, the price support for unconventional water from the local finance sector is selected.

Industrial water usage accounts for more than 90% of the total water usage, and all mine water is planned for industrial use; therefore, this paper conducts policy research on industrial water usage in the NECI Base. The Yellow River water price is 2.8 CNY/m³, and the mine water price is 10.8 CNY/m³ in the base year. Based on the allocation of the Yellow

River water and mine water in the planning year, the mine water subsidy scenarios are set from small to large, as shown in Table 5. Based on subsidizing the mine water until the price of the Yellow River water and mine water is equal (P6), the subsidy of mine water should increase and policy scenarios p7–p10 should be set, and the subsidy of mine water should be reduced and policy scenarios p1–p5 should be set. The non-subsidy scenario of mine water is the baseline scenario, and the subsidy scenarios of mine water are the policy scenarios.

**Table 5.** Mine water subsidy policy scenarios.

| Scenario Type | Scenario Number | Yellow River Water | | Mine Water | | |
|---|---|---|---|---|---|---|
| | | Price Increase Ratio Annual | Planning Year Water Price (CNY/m³) | Subsidy Amount | Planning Year Water Price (CNY/m³) | Price Decrease Ratio Annual |
| Baseline Scenario | P0 | 0% | 2.8 | 0 | 10.8 | 0% |
| Policy Scenarios | P1 | 1.00% | 2.9 | 0.8 | 10 | −1.50% |
| | P2 | 3.00% | 3.2 | 2.4 | 8.4 | −5.00% |
| | P3 | 5.00% | 3.6 | 4.2 | 6.6 | −9.50% |
| | P4 | 6.00% | 3.7 | 5.2 | 5.6 | −12.30% |
| | P5 | 7.00% | 3.9 | 6.2 | 4.6 | −15.60% |
| | P6 | 7.60% | 4 | 6.8 | 4 | −18.00% |
| | P7 | 8.00% | 4.1 | 7.2 | 3.6 | −19.80% |
| | P8 | 9.00% | 4.3 | 8.3 | 2.5 | −25.20% |
| | P9 | 10.00% | 4.5 | 9.4 | 1.4 | −33.40% |
| | P10 | 11.00% | 4.7 | 10.5 | 0.3 | −52.10% |

## 3. Results and Discussion

### 3.1. Results

The study simulates the mine water subsidy policy scenarios based on the CGE model of the extended WRM, and the IVA, the Yellow River water usage, and mine water usage are shown in Table 6.

**Table 6.** Water usage and IVA in different scenarios.

| Scenario Type | P0 | P1 | P2 | P3 | P4 | P5 | P6 | P7 | P8 | P9 | P10 |
|---|---|---|---|---|---|---|---|---|---|---|---|
| IVA (CNY Billion) | | | | | | | | | | | |
| Coal Mining and Chemical Industry | 40.32 | 40.31 | 40.31 | 40.31 | 40.31 | 40.31 | 40.31 | 40.31 | 40.32 | 40.33 | 40.35 |
| Thermal power Industry | 8.53 | 8.53 | 8.53 | 8.53 | 8.53 | 8.53 | 8.53 | 8.53 | 8.53 | 8.53 | 8.53 |
| New Materials Industry | 0.9 | 0.9 | 0.9 | 0.9 | 0.9 | 0.9 | 0.9 | 0.9 | 0.9 | 0.9 | 0.9 |
| Other Industries | 3.79 | 3.79 | 3.79 | 3.79 | 3.79 | 3.79 | 3.79 | 3.79 | 3.79 | 3.79 | 3.79 |
| The Total | 53.54 | 53.53 | 53.53 | 53.53 | 53.53 | 53.53 | 53.53 | 53.53 | 53.54 | 53.55 | 53.58 |
| The Yellow River Water Usage (Mm³) | | | | | | | | | | | |
| Coal Mining and Chemical Industry | 269.55 | 266.44 | 260.14 | 253.6 | 250.19 | 246.66 | 244.41 | 242.86 | 238.76 | 233.94 | 227.07 |
| Thermal power Industry | 42.83 | 41.84 | 39.82 | 37.7 | 36.57 | 35.39 | 34.61 | 34.07 | 32.63 | 30.83 | 28.02 |
| New Materials Industry | 1.61 | 1.61 | 1.61 | 1.61 | 1.61 | 1.61 | 1.61 | 1.61 | 1.61 | 1.61 | 1.61 |
| Other Industries | 5.03 | 5.03 | 5.03 | 5.03 | 5.03 | 5.03 | 5.03 | 5.03 | 5.03 | 5.03 | 5.04 |
| The Total | 319.03 | 314.93 | 306.6 | 297.94 | 293.41 | 288.69 | 285.67 | 283.58 | 278.03 | 271.42 | 261.74 |
| Mine Water Usage (Mm³) | | | | | | | | | | | |
| Coal Mining and Chemical Industry | 22.99 | 23.9 | 26.18 | 29.48 | 31.79 | 34.82 | 37.25 | 39.22 | 46.01 | 59.7 | 121.13 |
| Thermal power Industry | 4.88 | 5.07 | 5.52 | 6.15 | 6.58 | 7.14 | 7.58 | 7.94 | 9.15 | 11.57 | 22.26 |
| New Materials Industry | 0 | 0 | 0 | 0 | 0 | 0 | 0 | 0 | 0 | 0 | 0 |
| Other Industries | 0 | 0 | 0 | 0 | 0 | 0 | 0 | 0 | 0 | 0 | 0 |
| The Total | 27.88 | 28.97 | 31.69 | 35.63 | 38.37 | 41.96 | 44.83 | 47.15 | 55.17 | 71.26 | 143.39 |

### 3.2. Discussions

#### 3.2.1. Water Usage

With the increase in the mine water subsidy policy, the cost of using Yellow River water increases, the cost of using mine water decreases, the Yellow River water usage decreases, and the mine water usage increases. Referring to the allocation of Yellow River water and mine water in the planning year. When the Yellow River water price is raised by about 10.5% to subsidize the mine water, Yellow River water usage drops to 267.16 Mm³, as shown in Figure 3a. When the Yellow River water price is raised by about 8% to subsidize the mine water, mine water usage rises to 48.68 Mm³, as shown in Figure 3c.

With the increase in the mine water subsidy policy, the increase in mine water usage in the coal mining and chemical industry and thermal power industry shows an increasing trend, as shown in Figure 3d. The decrease in Yellow River usage in the coal mining and chemical industry and thermal power industry shows an increasing trend. The change in Yellow River usage in the material industry and other industries is not obvious, as shown in Figure 3b. There are two main reasons for the decrease in the Yellow River: (i) The main reason is that the increase in the mine water usage has replaced part of the Yellow River water, resulting in a decrease in Yellow River water usage. (ii) Raising the Yellow River price reduces Yellow River water usage owing to the water-saving effect of the raising price. However, the water-saving effect is limited due to the low elasticity of the water price. The new materials industry and other industries do not allocate mine water to replace Yellow River water, and they are water-saving industries with low water-saving potential. Therefore, the change in Yellow River water usage in the materials industry and other industries is not obvious.

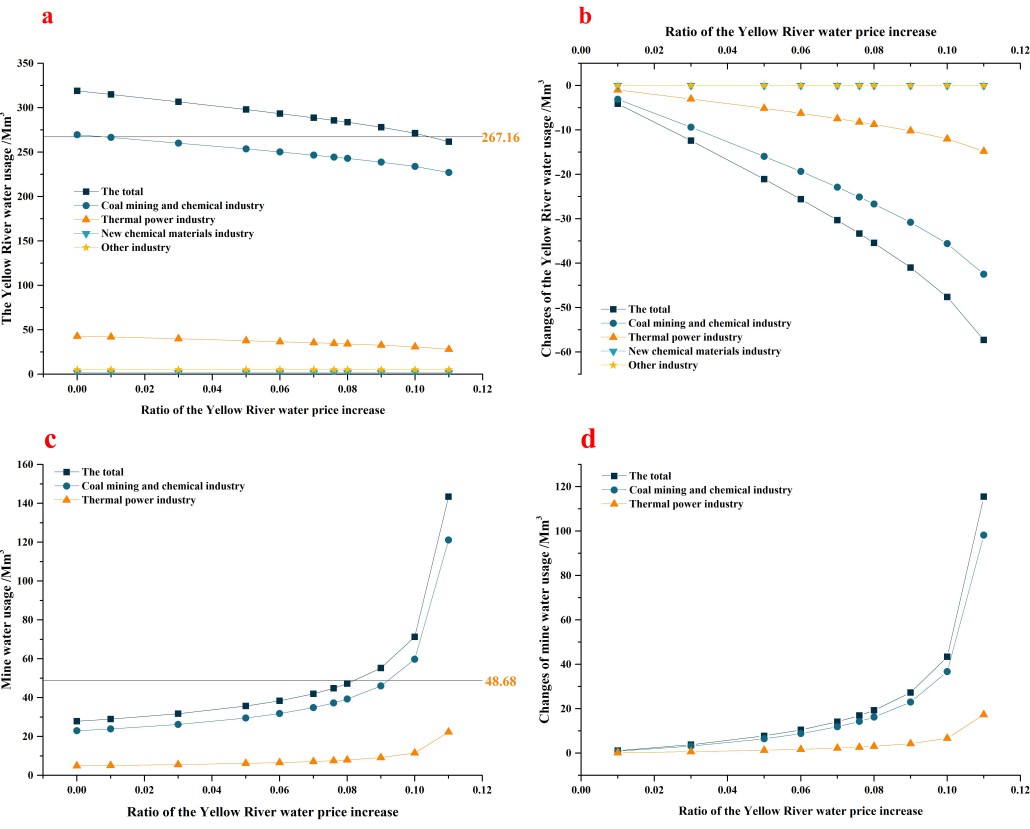

**Figure 3.** The Yellow River water usage and mine water usage and their changes compared with the baseline scenario in different scenarios: (**a**) description of the Yellow River water usage in different scenarios; (**b**) description of changes in the Yellow River water usage compared with the baseline scenario; (**c**) description of the mine water usage in different scenarios; (**d**) description of changes in the mine water compared with the baseline scenario.

### 3.2.2. IVA

Overall, with the increase in the mine water subsidy policy, the IVA does not change much, as shown in Figure 4a. It indicates that the policy has little impact on the macroeconomy of the NECI Base. This is because, although the mine water subsidy policy changes the enterprise water cost, the change in water cost is small compared to the IVA, so the IVA does not change much.

With the increase in the mine water subsidy policy, the decrease in IVA first increases and then decreases, and the IVA begins to increase in scenario P8, as shown in Figure 4b. When the mine water subsidy policy is small, the cost of mine water used by the enterprise is still much higher than that of Yellow River water. The enterprise still chooses to use Yellow River water. With the gradual increase in the mine water subsidy policy, the cost of water used by the enterprise increases, and the decrease in IVA gradually increases, as shown in Figure 4b (P1–P4). When the mine water subsidy policy is relatively large, with a gradual increase in the mine water subsidy policy, the enterprise tends to use mine water to reduce cost. The decrease in IVA begins to decrease, and the IVA begins to increase in scenario P8.

Compared to the change in IVA in different industries, the IVA of the coal mining and chemical industry has the most obvious change, as shown in Figure 4b. This is because the coal mining and chemical industry is a water-intensive industry, and it is also a major water user in the NECI Base. Thus, its IVA is significantly affected by water cost, while water usage in the thermal power industry, new material industry, and other industries is relatively small and thus not significantly affected by water cost. Therefore, their IVA has little impact.

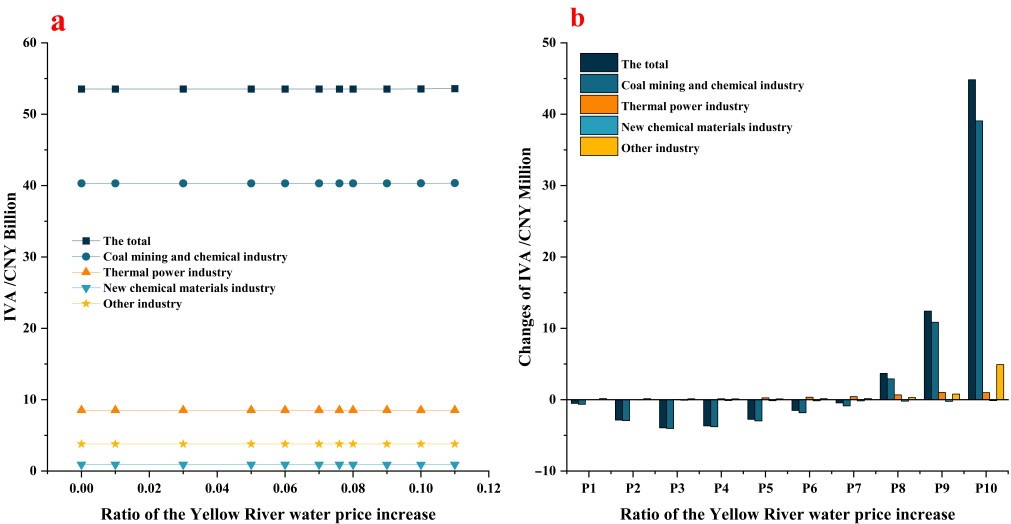

**Figure 4.** IVA and the change compared with the baseline scenario in different scenarios: (**a**) description of the IVA in different scenarios; (**b**) description of changes in the IVA compared with the baseline scenario.

### 3.2.3. WUE

The water usage per CNY 10,000 of IVA is taken as the indicator to analyze WUE, the water usage is the total of the Yellow River water and the mine water. The increase in water usage per CNY 10,000 of IVA indicates the decrease in WUE; otherwise, it indicates the increase. With the increase in the mine water subsidy policy, the WUE first increases and then decreases. When the Yellow River water price is increased by 8% to subsidize the mine water, which is the P7 scenario, the WUE is the highest, as shown in Figure 5a.

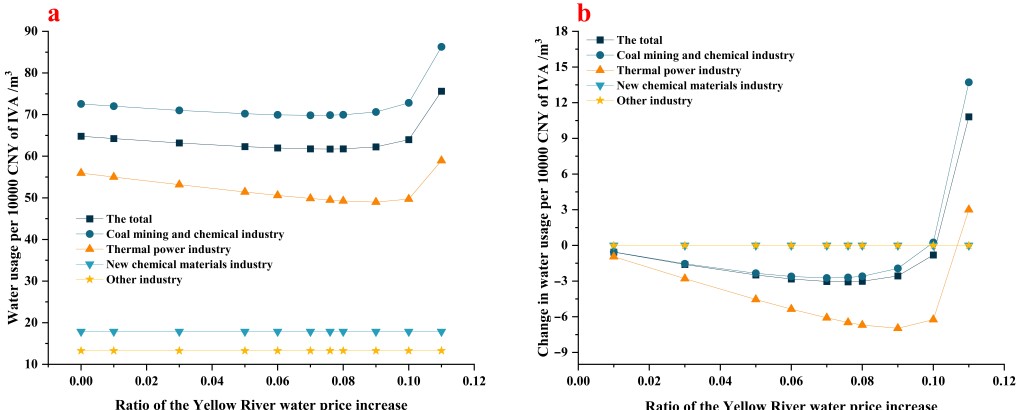

**Figure 5.** Water usage per CNY 10,000 of IVA and the change compared with the baseline scenario in different scenarios: (**a**) description of water usage per CNY 10,000 of IVA in different scenarios; (**b**) description of changes in water usage per CNY 10,000 of IVA compared with the baseline scenario.

With the increase in the mine water subsidy policy, the WUE of the coal mining and chemical industry and the thermal power industry shows a trend of rising first and then falling. However, the WUE of the new materials industry and other industries does not change much, as shown in Figure 5b. With the increase in the mine water subsidy policy, the increase in water cost makes water-intensive industries, such as the coal mining and chemical industry and the thermal power industry, save water and improves the WUE. When the Yellow River water price is increased by 8% to subsidize mine water, the cost of using mine water for enterprises is lower than the cost of using Yellow River water. As a result, enterprises use mine water instead of Yellow River water in large quantities to reduce the water cost. Therefore, a large amount of water is wasted, leading to the decrease in the WUE.

### 3.2.4. Comparison with Other Studies

Taking the NECI Base as the study area, this paper analyzes the impact of the mine water subsidy policy on the economy and water usage from the law and mechanism of policy action. These findings are consistent with previous research results: (i) In view of the impact of raising the water price on the economy and water usage, studies find that raising the water price leads to a decrease in IVA but has basically no impact on the macro-economy [49,50]. Studies find that raising water prices can encourage users to save water, but the water-saving effect is limited [51,52]. Studies have concluded that raising the water price can improve WUE, especially in water-intensive industries [49,51,53]. (ii) Studies have found that subsidizing unconventional water can promote unconventional water usage and replace conventional water in part [6,54–56]. (iii) Compared with water-saving industries, high water-intensive industries are more affected by water price policies, and the water price elasticity is greater [47].

### 3.2.5. Limitations and Directions

Overall, this paper improves the impact assessment mechanism of unconventional water utilization in the CGE model and provides a more complete quantitative analysis tool for unconventional water utilization policy research. However, there are limitations.

1.　Limited by the nature of the industry structure in the study area, this paper just discusses the impact of the mine water subsidy policy on the industry. It does not discuss the impact of the mine water subsidy policy on other industries. Therefore, choosing a study area with a more complete industry structure will make the study better.

2.　There are many policies to promote mine water utilization, such as the quota policy and the reward and punishment policy. This paper only studies the impact of the mine water subsidy policy on the economy and water usage. It does not study the

impact of other policies. Improving the model and analyzing the mechanisms and laws of other policies will be future research directions.

3. This paper does not discuss the ecological value of unconventional water utilization, improving the model and analyzing the ecological value of unconventional water utilization will be future research directions.

4. The good operation of the water network can guarantee reliable and safe delivery to consumers, and the analysis of water network failure can guarantee the water network's good operation. The analysis and prediction of water distribution network failures and deterioration through the prediction model will be included in future work [57].

## 4. Conclusions and Recommendations

### 4.1. Conclusions

In order to solve the shortage of water resources and achieve high-quality regional development, making full use of unconventional water resources is one of the key solutions. However, the high cost of unconventional water prevents its further utilization. Taking the NECI Base as an example, this paper proposes that by raising the price of Yellow River water to subsidize the utilization of mine water, the development and utilization of unconventional water can be effectively improved. Faced with the lack of quantitative analysis of the policy in current research, this paper extends the WRM of the CGE model. Through the analysis of ten different mine water subsidy scenarios, the conclusions are as follows:

1. This paper extends the WRM and incorporates it into the CGE model by means of the intermediate inputs. The extended CGE model can not only reflect the substitution relationship between Yellow River water and mine water but also the optimal allocation of multiple water sources in the NECI Base. The extended CGE model can realize the quantitative analysis of unconventional water utilization and reveal the mechanism and laws of the mine water subsidy policy.

2. Through the model, this paper simulates the mine water subsidy policy. The law and mechanisms of the mine water subsidy policy are revealed through quantitative analysis. (i) With the increase in the mine water subsidy policy, Yellow River water usage decreases, and mine water usage increases, which indicates that the mine water subsidy policy can optimize the water-usage structure. The change in IVA is not obvious, indicating that the mine water subsidy policy has little impact on the macro-economy of the NECI Base. The WUE shows a trend of rising first and then falling, which indicates that a reasonable price difference should be established between Yellow River water and mine water to ensure the efficient utilization of water resources. (ii) Water-intensive industries, such as the coal mining and chemical industry and the thermal power industry, are greatly affected by the mine water subsidy policy. However, water-saving industries, such as the new materials industry, and other industries are less affected by the mine water subsidy policy.

3. According to the law of the mine water subsidy policy and the water resource allocation target, it is recommended to increase the price of Yellow River water by 8% to subsidize mine water. By 2025, the Yellow River water price is 4.1 CNY/m³, and the mine water price is 3.6 CNY/m³; both of them do not exceed the cost of 8.78 CNY/m³. Under this scenario, the IVA is basically unaffected, the WUE is significantly improved, and the affordability of the enterprise is satisfied. Yellow River water usage decreases from 319.03 Mm³ to 283.58 Mm³ (11.1% saving), and Yellow River water usage increases from 27.88 Mm³ to 47.15 Mm³ (69.1% increase). These basically reach the goal of allocating Yellow River water and mine water in the NECI Base.

### 4.2. Recommendations

It is recommended to increase the price of Yellow River water by 8% to subsidize mine water. To successfully implement the mine water subsidy policy, some recommendations may apply to authorities:

1.  Establish special coordination and management institutions for the development and utilization of unconventional water resources, clarify the responsibility boundary between government departments in the NECI Base, and set up an inter-departmental coordination mechanism. In this way, unified coordination, planning, and management of Yellow River and water mine water can be achieved.

2.  Establish the supervision, examination, and publicity system for the water supply cost of Yellow River water and mine water and dynamically assess the change in the water supply cost and affordability. According to the scheme of increasing Yellow River water's price by 8% to subsidize mine water, the prices of Yellow River water and mine water are dynamically adjusted.

Taking the NECI Base as an example, this paper quantitatively studies the subsidy policy of unconventional water based on the CGE model, hoping to provide a decision-making reference for the utilization and policy formulation of unconventional water in China.

**Author Contributions:** Conceptualization, C.M. and H.N.; methodology, C.M., H.N., Y.J. and X.L.; software, C.M., H.N. and Y.J.; formal analysis, C.M. and H.N.; validation, C.M.; investigation, C.M., H.N. and X.L.; data curation, C.M. and X.L.; writing—original draft preparation, C.M.; writing—review and editing, C.M. and H.N.; visualization, C.M. and H.N.; supervision, H.N. and Y.J.; project administration, H.N.; funding acquisition, Y.J. All authors have read and agreed to the published version of the manuscript.

**Funding:** This research was funded by the National Key Research and Development Plan OF CHINA grant number 2021YFC3200204.

**Institutional Review Board Statement:** Not applicable.

**Informed Consent Statement:** Not applicable.

**Data Availability Statement:** The data presented in this study are available on request from the corresponding author.

**Conflicts of Interest:** The authors declare no conflict of interest.

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
