# Peer review of "Study on the Unconventional Water Subsidy Policy in the Arid Area of Northwest China"

_water, doi:10.3390/w14193167_

Round 1

Reviewer 1 Report

The manuscript entitled “Study on unconventional water subsidy policy in the arid area of Northwest China” contains interesting research. The authors attempt to seek the impact of various pricing and subsidy regimes on optimal allocation of water. The CGE model is used for this analysis. The results are interesting and have to potential to inform policy toward optimal allocation of water. The paper is well-structured, and most parts of the manuscript are described clearly. I have few comments and suggestions for the authors:

Please add an extended literature review section and discuss the studies and methods aimed at investigating the optimal allocation of water both at home and abroad. Some studies are suggested below:

Razzaq et al. 2019. Can the informal groundwater markets improve water use efficiency and equity? Evidence from a semi-arid region of Pakistan. Science of The Total Environment, 666, 849-857.

Liu et al., 2010. A model for the optimal allocation of water resources in a saltwater intrusion area: a case study in Pearl River Delta in China. Water resources management, 24(1), pp.63-81.

Razzaq et al., 2022. The Competitiveness, Bargaining Power, and Contract Choice in Agricultural Water Markets in Pakistan: Implications for Price Discrimination and Environmental Sustainability. Frontiers in Environmental Science, 10.

The paper needs to be revised to improve the language at some places. Some of the sentences are too long and complex.

On page 6, both headings are named as Production Module.

Please enhance the discussion section.

Discuss the limitations in length of your research and add future research directions.

Reviewer 2 Report

In the study, the Authors focused on the crucial issue of unconventional water subsidy policy in the arid area of Northwest China. Ten different subsidy policy scenarios were simulated through the extended CGE model, and the laws and mechanisms of subsidy policy on the economy and water usage are summarized. Results show that increasing the price of Yellow River water by 8% to subsidize the mine water will achieve optimal socio-economic output. Under this scenario, the industrial added value was basically unaffected, while the water efficiency was significantly improved. Remarks: Add reference to the equations in case you are not the Author of them. Are there concrete steps that can be recommended for the authorities? The advantages and novelty of the research approach need to add. Please, make a reference to this issue. This will help highlight any unique findings.

Reviewer 3 Report

The paper covers an important topic and the policy implications can be very useful for other countries. The authors analysed the unconventional water subsidy policy in the arid area of Northwest China and proposed that by raising the Yellow River water price to subsidize the utilization of mine water, the development and utilization of mine water can be effectively improved. The paper has potential although several aspects must be taken into account by the authors before being considered for publication, namely:

·       The keywords can be revised, for example it could be separated the word “Subsidy policy”

·       The introduction can also be improved. Tariff setting must follow several aspects (see Pinto and Marques, 2017);

·       The authors should also highlight the novelty of the paper in the introduction;

·       It would be also important to add a paragraph presenting the organization of the paper;

·       The literature review must be strongly improved;

·       Regarding the methodology, the model must be better justified, including the limitations;

·       All sources of information must de detailed in the text;

·       Highlight the importance and main aspects of the different subsidies (see Narzetti and Marques, 2020);

·       All the abbreviations must be presented in the text;

·       How regulation can contribute to improve the performance in this regard?;

·       Affordability is also a relevant matter. How this matter must be taken into account? There is reliable data to evaluate this aspect?;

·       More recommendations for decision makers were expected in the discussion and conclusions;

·       The references must be homogenized and in line with the guidelines (for example some issues are missing in the references).

References

PINTO, F.; MARQUES, R. (2017). New Era / New Solutions: the role of alternative tariff structures in water supply projects. Water Research. Elsevier. ISSN: 0043-1354. Vol. 126, pp.  216-231.

NARZETTI, D.; MARQUES, R. (2020). Models of subsidies for water and sanitation services for vulnerable people in South American countries: lessons for Brazil. Water. ISSN: 2073-4441. Vol. 12 (7), 1976.

Round 2

Reviewer 2 Report

In the study, the Authors focused on the crucial issue of unconventional water subsidy policy in the arid area of Northwest China. Ten different subsidy policy scenarios were simulated through the extended CGE model, and the laws and mechanisms of subsidy policy on the economy and water usage are summarized. Results show that increasing the price of Yellow River water by 8% to subsidize the mine water will achieve optimal socio-economic output. Under this scenario, the industrial added value was basically unaffected, while the water efficiency was significantly improved. Remarks: Line 391: The choice of reference should be supplemented with respect to the studies have found that subsidizing unconventional water can promote unconventional water usage and replace part of conventional water and future research of using the prediction model ( e.g. Ref. Modelling water distribution network failures and deterioration, 2017, IEEE International Conference on Industrial Engineering and Engineering Management 2017-December, 924-928. DOI 10.1109/IEEM.2017.8290027. Add some perspectives of the future work.
